# Decomposing Parameter Estimation Problems

**Khaled S. Refaat, Arthur Choi, Adnan Darwiche**
Computer Science Department
University of California, Los Angeles
{krefaat,aychoi,darwiche}@cs.ucla.edu

## Abstract

We propose a technique for decomposing the parameter learning problem in Bayesian networks into independent learning problems. Our technique applies to incomplete datasets and exploits variables that are either hidden or observed in the given dataset. We show empirically that the proposed technique can lead to orders-of-magnitude savings in learning time. We explain, analytically and empirically, the reasons behind our reported savings, and compare the proposed technique to related ones that are sometimes used by inference algorithms.

## 1 Introduction

Learning Bayesian network parameters is the problem of estimating the parameters of a known structure given a dataset. This learning task is usually formulated as an optimization problem that seeks maximum likelihood parameters: ones that maximize the probability of a dataset.

A key distinction is commonly drawn between complete and incomplete datasets. In a complete dataset, the value of each variable is known in every example. In this case, maximum likelihood parameters are unique and can be easily estimated using a single pass on the dataset. However, when the data is incomplete, the optimization problem is generally non-convex, has multiple local optima, and is commonly solved by iterative methods, such as EM [5, 7], gradient descent [13] and, more recently, EDML [2, 11, 12].

Incomplete datasets may still exhibit a certain structure. In particular, certain variables may always be observed in the dataset, while others may always be unobserved (hidden). We exploit this structure by decomposing the parameter learning problem into smaller learning problems that can be solved independently. In particular, we show that the stationary points of the likelihood function can be characterized by the ones of the smaller problems. This implies that algorithms such as EM and gradient descent can be applied to the smaller problems while preserving their guarantees. Empirically, we show that the proposed decomposition technique can lead to orders-of-magnitude savings. Moreover, we show that the savings are amplified when the dataset grows in size. Finally, we explain these significant savings analytically by examining the impact of our decomposition technique on the dynamics of the used convergence test, and on the properties of the datasets associated with the smaller learning problems.

The paper is organized as follows. In Section 2, we provide some background on learning Bayesian network parameters. In Section 3, we present the decomposition technique and then prove its soundness in Section 4. Section 5 is dedicated to empirical results and to analyzing the reported savings. We discuss related work in Section 6 and finally close with some concluding remarks in Section 7. The proofs are moved to the appendix in the supplementary material.

## 2 Learning Bayesian Network Parameters

We use upper case letters $(X)$ to denote variables and lower case letters $(x)$ to denote their values. Variable sets are denoted by bold-face upper case letters $(\mathbf{X})$ and their instantiations by bold-face lower case letters $(\mathbf{x})$. Generally, we will use $X$ to denote a variable in a Bayesian network and $\mathbf{U}$ to denote its parents.

A Bayesian network is a directed acyclic graph with a conditional probability table (CPT) associated with each node $X$ and its parents $\mathbf{U}$. For every variable instantiation $x$ and parent instantiation $\mathbf{u}$, the CPT of $X$ includes a parameter $\theta_{x|\mathbf{u}}$ that represents the probability $Pr(X\!=\!x|\mathbf{U}\!=\!\mathbf{u})$. We will use $\theta$ to denote the set of all network parameters. Parameter learning in Bayesian networks is the process of estimating these parameters $\theta$ from a given dataset.

A *dataset* is a multi-set of *examples.* Each example is an instantiation of some network variables. We will use $\mathcal{D}$ to denote a dataset and $\mathbf{d}_1, \ldots, \mathbf{d}_N$ to denote its $N$ examples. The following is a dataset over four binary variables ("?" indicates a missing value of a variable in an example):

| example | $E$ | $B$ | $A$ | $C$ |
|---------|-----|-----|-----|-----|
| $\mathbf{d}_1$ | $e$ | $b$ | $a$ | ? |
| $\mathbf{d}_2$ | ? | $\bar{b}$ | $\bar{a}$ | ? |
| $\mathbf{d}_3$ | $e$ | $\bar{b}$ | $\bar{a}$ | ? |

A variable $X$ is *observed* in a dataset iff the value of $X$ is known in each example of the dataset (i.e., "?" cannot appear in the column corresponding to variable $X$). Variables $A$ and $B$ are observed in the above dataset. Moreover, a variable $X$ is *hidden* in a dataset iff its value is unknown in every example of the dataset (i.e., only "?" appears in the column of variable $X$). Variable $C$ is hidden in the above dataset. When all variables are observed in a dataset, the dataset is said to be *complete.* Otherwise, the dataset is *incomplete.* The above dataset is incomplete.

Given a dataset $\mathcal{D}$ with examples $\mathbf{d}_1, \ldots, \mathbf{d}_N$, the *likelihood* of parameter estimates $\theta$ is defined as:

$$L(\theta|\mathcal{D}) = \prod_{i=1}^{N} Pr_{\theta}(\mathbf{d}_i).$$

Here, $Pr_{\theta}$ is the distribution induced by the network structure and parameters $\theta$. One typically seeks maximum likelihood parameters

$$\theta^{\star} = \underset{\theta}{\operatorname{argmax}}\, L(\theta|\mathcal{D}).$$

When the dataset is complete, maximum likelihood estimates are unique and easily obtainable using a single pass over the dataset (e.g., [3, 6]). For incomplete datasets, the problem is generally non-convex and has multiple local optima. Iterative algorithms are usually used in this case to try to obtain maximum likelihood estimates. This includes EM [5, 7], gradient descent [13], and the more recent EDML algorithm [2, 11, 12]. The fixed points of these algorithms correspond to the stationary points of the likelihood function. Hence, these algorithms are not guaranteed to converge to global optima. As such, they are typically applied to multiple seeds (initial parameter estimates), while retaining the best estimates obtained across all seeds.

## 3 Decomposing the Learning Problem

We now show how the problem of learning Bayesian network parameters can be decomposed into independent learning problems. The proposed technique exploits two aspects of a dataset: hidden and observed variables.

**Proposition 1** *The likelihood function $L(\theta|\mathcal{D})$ does not depend on the parameters of variable $X$ if $X$ is hidden in dataset $\mathcal{D}$ and is a leaf of the network structure.*

If a hidden variable appears as a leaf in the network structure, it can be removed from the structure while setting its parameters arbitrarily (assuming no prior). This process can be repeated until there are no leaf variables that are also hidden. The soundness of this technique follows from [14, 15].

Our second decomposition technique will exploit the observed variables of a dataset. In a nutshell, we will (a) decompose the Bayesian network into a number of sub-networks, (b) learn the parameters of each sub-network independently, and then (c) assemble parameter estimates for the original network from the estimates obtained in each sub-network.

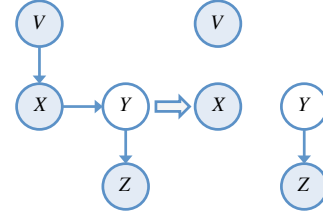

Figure 1: Identifying components of network $G$ given $\mathbf{O} = \{V, X, Z\}$.

**Definition 1 (Component)** *Let $G$ be a network, $\mathbf{O}$ be some observed variables in $G$ and let $G|\mathbf{O}$ be the network which results from deleting all edges from $G$ which are outgoing from $\mathbf{O}$. A component of $G|\mathbf{O}$ is a maximal set of nodes that are connected in $G|\mathbf{O}$.*

Consider the network $G$ in Figure 1, with observed variables $\mathbf{O} = \{V, X, Z\}$. Then $G|\mathbf{O}$ has three components in this case: $\mathbf{S}_1 = \{V\}$, $\mathbf{S}_2 = \{X\}$, and $\mathbf{S}_3 = \{Y, Z\}$.

The components of a network partition its parameters into groups, one group per component. In the above example, the network parameters are partitioned into the following groups:

$$
\begin{aligned}
\mathbf{S}_1 : &\quad \{\theta_v, \theta_{\overline{v}}\} \\
\mathbf{S}_2 : &\quad \{\theta_{x|v}, \theta_{\overline{x}|v}, \theta_{x|\overline{v}}, \theta_{\overline{x}|\overline{v}}\} \\
\mathbf{S}_3 : &\quad \{\theta_{y|x}, \theta_{\overline{y}|x}, \theta_{y|\overline{x}}, \theta_{\overline{y}|\overline{x}}, \theta_{z|y}, \theta_{\overline{z}|y}, \theta_{z|\overline{y}}, \theta_{\overline{z}|\overline{y}}\}.
\end{aligned}
$$

We will later show that the learning problem can be decomposed into independent learning problems, each induced by one component. To define these independent problems, we need some definitions.

**Definition 2 (Boundary Node)** *Let $\mathbf{S}$ be a component of $G|\mathbf{O}$. If edge $B \to S$ appears in $G$, $B \notin \mathbf{S}$ and $S \in \mathbf{S}$, then $B$ is called a boundary for component $\mathbf{S}$.*

Considering Figure 1, node $X$ is the only boundary for component $\mathbf{S}_3 = \{Y, Z\}$. Moreover, node $V$ is the only boundary for component $\mathbf{S}_2 = \{X\}$. Component $\mathbf{S}_1 = \{V\}$ has no boundary nodes.

The independent learning problems are based on the following sub-networks.

**Definition 3 (Sub-Network)** *Let $\mathbf{S}$ be a component of $G|\mathbf{O}$ with boundary variables $\mathbf{B}$. The sub-network of component $\mathbf{S}$ is the subset of network $G$ induced by variables $\mathbf{S} \cup \mathbf{B}$.*

Figure 2 depicts the three sub-networks which correspond to our running example.

The parameters of a sub-network will be learned using projected datasets.

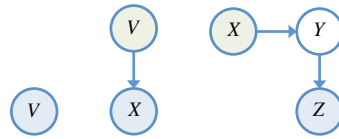

**Definition 4** *Let $\mathcal{D} = \mathbf{d}_1, \dots, \mathbf{d}_N$ be a dataset over variables $\mathbf{X}$ and let $\mathbf{Y}$ be a subset of variables $\mathbf{X}$. The projection of dataset $\mathcal{D}$ on variables $\mathbf{Y}$ is the set of examples $\mathbf{e}_1, \dots, \mathbf{e}_N$, where each $\mathbf{e}_i$ is the subset of example $\mathbf{d}_i$ which pertains to variables $\mathbf{Y}$.*

Figure 2: The sub-networks induced by adding boundary variables to components.

We show below a dataset for the full Bayesian network in Figure 1, followed by three projected datasets, one for each of the sub-networks in Figure 2.

| | $V$ | $X$ | $Y$ | $Z$ |
|---|---|---|---|---|
| $\mathbf{d}_1$ | $\overline{v}$ | $x$ | ? | $\overline{z}$ |
| $\mathbf{d}_2$ | $v$ | $\overline{x}$ | ? | $z$ |
| $\mathbf{d}_3$ | $v$ | $x$ | ? | $\overline{z}$ |

| | $V$ | count |
|---|---|---|
| $\mathbf{e}_1$ | $\overline{v}$ | 1 |
| $\mathbf{e}_2$ | $v$ | 2 |

| | $V$ | $X$ | count |
|---|---|---|---|
| $\mathbf{e}_1$ | $\overline{v}$ | $x$ | 1 |
| $\mathbf{e}_2$ | $v$ | $\overline{x}$ | 1 |
| $\mathbf{e}_3$ | $v$ | $x$ | 1 |

| | $X$ | $Y$ | $Z$ | count |
|---|---|---|---|---|
| $\mathbf{e}_1$ | $x$ | ? | $\overline{z}$ | 2 |
| $\mathbf{e}_2$ | $\overline{x}$ | ? | $z$ | 1 |

The projected datasets are "compressed" as we only represent unique examples, together with a count of how many times each example appears in a dataset. Using compressed datasets is crucial to realizing the full potential of decomposition, as it ensures that the size of a projected dataset is at most exponential in the number of variables appearing in its sub-network (more on this later).

We are now ready to describe our decomposition technique. Given a Bayesian network structure $G$ and a dataset $\mathcal{D}$ that observes variables $\mathbf{O}$, we can get the stationary points of the likelihood function for network $G$ as follows:

1. Identify the components $\mathbf{S}_1, \ldots, \mathbf{S}_M$ of $G|\mathbf{O}$ (Definition 1).
2. Construct a sub-network for each component $\mathbf{S}_i$ and its boundary variables $\mathbf{B}_i$ (Definition 3).
3. Project the dataset $\mathcal{D}$ on the variables of each sub-network (Definition 4).
4. Identify a stationary point for each sub-network and its projected dataset (using, e.g., EM, EDML or gradient descent).
5. Recover the learned parameters *of non-boundary variables* from each sub-network.

We will next prove that (a) these parameters are a stationary point of the likelihood function for network $G$, and (b) every stationary point of the likelihood function can be generated this way (using an appropriate seed).

## 4 Soundness

The soundness of our decomposition technique is based on three steps. We first introduce the notion of a *parameter term,* on which our proof rests. We then show how the likelihood function for the Bayesian network can be decomposed into component likelihood functions, one for each sub-network. We finally show that the stationary points of the likelihood function (network) can be characterized by the stationary points of component likelihood functions (sub-networks).

Two parameters are *compatible* iff they agree on the state of their common variables. For example, parameters $\theta_{z|y}$ and $\theta_{y|x}$ are compatible, but parameters $\theta_{z|\overline{y}}$ and $\theta_{y|x}$ are not compatible, as $y \neq \overline{y}$. Moreover, a parameter is compatible with an example iff they agree on the state of their common variables. Parameter $\theta_{\overline{y}|x}$ is compatible with example $x, \overline{y}, z$, but not with example $x, y, z$.

**Definition 5 (Parameter Term)** *Let* $\mathbf{S}$ *be network variables and let* $\mathbf{d}$ *be an example. A* *parameter term for* $\mathbf{S}$ *and* $\mathbf{d}$, *denoted* $\Theta_{\mathbf{S}}^{\mathbf{d}}$, *is a product of compatible network parameters, one for each variable in* $\mathbf{S}$, *that are also compatible with example* $\mathbf{d}$.

Consider the network $X \to Y \to Z$. If $\mathbf{S} = \{Y, Z\}$ and $\mathbf{d} = x, z$, then $\Theta_{\mathbf{S}}^{\mathbf{d}}$ will denote either $\theta_{y|x} \theta_{z|y}$ or $\theta_{\overline{y}|x} \theta_{z|\overline{y}}$. Moreover, if $\mathbf{S} = \{X, Y, Z\}$, then $\Theta_{\mathbf{S}}^{\mathbf{d}}$ will denote either $\theta_x \theta_{y|x} \theta_{z|y}$ or $\theta_x \theta_{\overline{y}|x} \theta_{z|\overline{y}}$. In this case, $Pr(\mathbf{d}) = \sum_{\Theta_{\mathbf{S}}^{\mathbf{d}}} \Theta_{\mathbf{S}}^{\mathbf{d}}$. This holds more generally, whenever $\mathbf{S}$ is the set of all network variables.

We will now use parameter terms to show how the likelihood function can be decomposed into component likelihood functions.

**Theorem 1** *Let* $\mathbf{S}$ *be a component of* $G|\mathbf{O}$ *and let* $\mathbf{R}$ *be the remaining variables of network* $G$. *If variables* $\mathbf{O}$ *are observed in example* $\mathbf{d}$, *we have*

$$Pr_{\theta}(\mathbf{d}) = \left[\sum_{\Theta_{\mathbf{S}}^{\mathbf{d}}} \Theta_{\mathbf{S}}^{\mathbf{d}}\right] \left[\sum_{\Theta_{\mathbf{R}}^{\mathbf{d}}} \Theta_{\mathbf{R}}^{\mathbf{d}}\right].$$

If $\theta$ denotes all network parameters, and $\mathbf{S}$ is a set of network variables, then $\theta : \mathbf{S}$ will denote the subset of network parameters which pertain to the variables in $\mathbf{S}$. Each component $\mathbf{S}$ of a Bayesian network induces its own likelihood function over parameters $\theta : \mathbf{S}$.

**Definition 6 (Component Likelihood)** *Let* $\mathbf{S}$ *be a component of* $G|\mathbf{O}$. *For dataset* $\mathcal{D} = \mathbf{d}_1, \ldots, \mathbf{d}_N$, *the component likelihood for* $\mathbf{S}$ *is defined as*

$$L(\theta : \mathbf{S} | \mathcal{D}) = \prod_{i=1}^{N} \sum_{\Theta_{\mathbf{S}}^{\mathbf{d}_i}} \Theta_{\mathbf{S}}^{\mathbf{d}_i}.$$

In our running example, the components are $\mathbf{S}_1 = \{V\}$, $\mathbf{S}_2 = \{X\}$ and $\mathbf{S}_3 = \{Y, Z\}$. Moreover, the observed variables are $\mathbf{O} = \{V, X, Z\}$. Hence, the component likelihoods are

$$
\begin{aligned}
L(\theta : \mathbf{S}_1 | \mathcal{D}) &= \left[\theta_{\overline{v}}\right] \left[\theta_v\right] \left[\theta_v\right] \\
L(\theta : \mathbf{S}_2 | \mathcal{D}) &= \left[\theta_{x|\overline{v}}\right] \left[\theta_{\overline{x}|v}\right] \left[\theta_{x|v}\right] \\
L(\theta : \mathbf{S}_3 | \mathcal{D}) &= \left[\theta_{y|x}\theta_{\overline{z}|y} + \theta_{\overline{y}|x}\theta_{\overline{z}|\overline{y}}\right] \left[\theta_{y|\overline{x}}\theta_{z|y} + \theta_{\overline{y}|\overline{x}}\theta_{z|\overline{y}}\right] \left[\theta_{y|x}\theta_{\overline{z}|y} + \theta_{\overline{y}|x}\theta_{\overline{z}|\overline{y}}\right]
\end{aligned}
$$

The parameters of component likelihoods partition the network parameters. That is, the parameters of two component likelihoods are always non-overlapping. Moreover, the parameters of component likelihoods account for all network parameters.[1]

We can now state our main decomposition result, which is a direct corollary of Theorem 1.

**Corollary 1** *Let* $\mathbf{S}_1, \ldots, \mathbf{S}_M$ *be the components of* $G|\mathbf{O}$. *If variables* $\mathbf{O}$ *are observed in dataset* $\mathcal{D}$,

$$
L(\theta | \mathcal{D}) = \prod_{i=1}^{M} L(\theta : \mathbf{S}_i | \mathcal{D}).
$$

Hence, the network likelihood decomposes into a product of component likelihoods. This leads to another important corollary (see Lemma 1 in the Appendix):

**Corollary 2** *Let* $\mathbf{S}_1, \ldots, \mathbf{S}_M$ *be the components of* $G|\mathbf{O}$. *If variables* $\mathbf{O}$ *are observed in dataset* $\mathcal{D}$, *then* $\theta^\star$ *is a stationary point of the likelihood* $L(\theta|\mathcal{D})$ *iff, for each* $i$, $\theta^\star : \mathbf{S}_i$ *is a stationary point for the component likelihood* $L(\theta : \mathbf{S}_i | \mathcal{D})$.

The search for stationary points of the network likelihood is now decomposed into independent searches for stationary points of component likelihoods.

We will now show that the stationary points of a component likelihood can be identified using any algorithm that identifies such points for the network likelihood.

**Theorem 2** *Consider a sub-network* $G$ *which is induced by component* $\mathbf{S}$ *and boundary variables* $\mathbf{B}$. *Let* $\theta$ *be the parameters of sub-network* $G$, *and let* $\mathcal{D}$ *be a dataset for* $G$ *that observes boundary variables* $\mathbf{B}$. *Then* $\theta^\star$ *is a stationary point for the sub-network likelihood,* $L(\theta|\mathcal{D})$, *only if* $\theta^\star : \mathbf{S}$ *is a stationary point for the component likelihood* $L(\theta : \mathbf{S} | \mathcal{D})$. *Moreover, every stationary point for* $L(\theta : \mathbf{S} | \mathcal{D})$ *is part of some stationary point for* $L(\theta|\mathcal{D})$.

Given an algorithm that identifies stationary points of the likelihood function of Bayesian networks (e.g., EM), we can now identify all stationary points of a component likelihood. That is, we just apply this algorithm to the sub-network of each component $\mathbf{S}$, and then extract the parameter estimates of variables in $\mathbf{S}$ while *ignoring* the parameters of boundary variables. This proves the soundness of our proposed decomposition technique.

## 5    The Computational Benefit of Decomposition

We will now illustrate the computational benefits of the proposed decomposition technique, showing orders-of-magnitude reductions in learning time. Our experiments are structured as follows. Given a Bayesian network $G$, we generate a dataset $\mathcal{D}$ while ensuring that a certain percentage of variables are observed, with all others hidden. Using dataset $\mathcal{D}$, we estimate the parameters of network $G$ using two methods. The first uses the classical EM on network $G$ and dataset $\mathcal{D}$. The second decomposes network $G$ into its sub-networks $G_1, \ldots, G_M$, projects the dataset $\mathcal{D}$ on each sub-network, and then applies EM to each sub-network and its projected dataset. This method is called D-EM (for Decomposed EM). We use the same seed for both EM and D-EM.

Before we present our results, we have the following observations on our data generation model. First, we made all unobserved variables hidden (as opposed to missing at random) as this leads to a more difficult learning problem, especially for EM (even with the pruning of hidden leaf nodes).

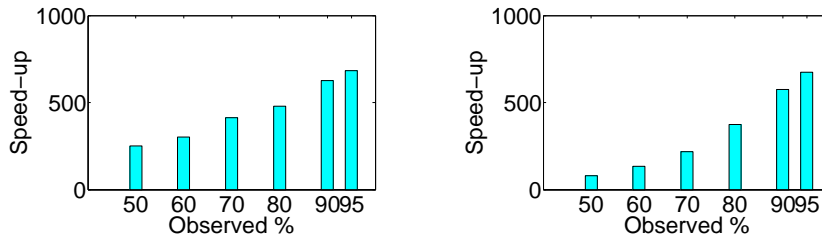

Figure 3: Speed-up of D-EM over EM on chain networks: three chains (180, 380, and 500 variables) (left), and tree networks (63, 127, 255, and 511 variables) (right), with three random datasets per network/observed percentage, and $2^{10}$ examples per dataset.

| Observed % | Network | Speed-up D-EM | Network | Speed-up D-EM | Network | Speed-up D-EM |
|---|---|---|---|---|---|---|
| 95.0% | alarm | 267.67x | diagnose | 43.03x | andes | 155.54x |
| 90.0% | alarm | 173.47x | diagnose | 17.16x | andes | 52.63x |
| 80.0% | alarm | 115.4x | diagnose | 11.86x | andes | 14.27x |
| 70.0% | alarm | 87.67x | diagnose | 3.25x | andes | 2.96x |
| 60.0% | alarm | 92.65x | diagnose | 3.48x | andes | 0.77x |
| 50.0% | alarm | 12.09x | diagnose | 3.73x | andes | 1.01x |
| 95.0% | win95pts | 591.38x | water | 811.48x | pigs | 235.63x |
| 90.0% | win95pts | 112.57x | water | 110.27x | pigs | 37.61x |
| 80.0% | win95pts | 22.41x | water | 7.23x | pigs | 34.19x |
| 70.0% | win95pts | 17.92x | water | 1.5x | pigs | 16.23x |
| 60.0% | win95pts | 4.8x | water | 2.03x | pigs | 4.1x |
| 50.0% | win95pts | 7.99x | water | 4.4x | pigs | 3.16x |

Table 1: Speed-up of D-EM over EM on UAI networks. Three random datasets per network/observed percentage with $2^{10}$ examples per dataset.

Second, it is not uncommon to have a significant number of variables that are always observed in real-world datasets. For example, in the UCI repository: the internet advertisements dataset has 1558 variables, only 3 of which have missing values; the automobile dataset has 26 variables, where 7 have missing values; the dermatology dataset has 34 variables, where only age can be missing; and the mushroom dataset has 22 variables, where only one variable has missing values [1].

We performed our experiments on three sets of networks: synthesized chains, synthesized complete binary trees, and some benchmarks from the UAI 2008 evaluation with other standard benchmarks (called UAI networks): alarm, win95pts, andes, diagnose, water, and pigs. Figure 3 and Table 1 depict the obtained time savings. As can be seen from these results, decomposing chains and trees lead to two orders-of-magnitude speed-ups for almost all observed percentages. For UAI networks, when observing 70% of the variables or more, one obtains one-to-two orders-of-magnitude speed-ups. We note here that the time used for D-EM includes the time needed for decomposition (i.e., identifying the sub-networks and their projected datasets). Similar results for EDML are shown in the supplementary material.

The reported computational savings appear quite surprising. We now shed some light on the culprit behind these savings. We also argue that some of the most prominent tools for Bayesian networks do not appear to employ the proposed decomposition technique when learning network parameters.

**Our first analytic explanation for the obtained savings is based on understanding the role of data projection**, which can be illustrated by the following example. Consider a chain network over binary variables $X_1, \ldots, X_n$, where $n$ is even. Consider also a dataset $\mathcal{D}$ in which variable $X_i$ is observed for all odd $i$. There are $n/2$ sub-networks in this case. The first sub-network is $X_1$. The remaining sub-networks are in the form $X_{i-1} \to X_i \to X_{i+1}$ for $i = 2, 4, \ldots, n - 2$ (node $X_n$ will be pruned). The dataset $\mathcal{D}$ can have up to $2^{n/2}$ distinct examples. If one learns parameters without decomposition, one would need to call the inference engine once for each distinct example, in each iteration of the learning algorithm. With $m$ iterations, the inference engine may be called up to $m2^{n/2}$ times. When learning with decomposition, however, each projected dataset will have

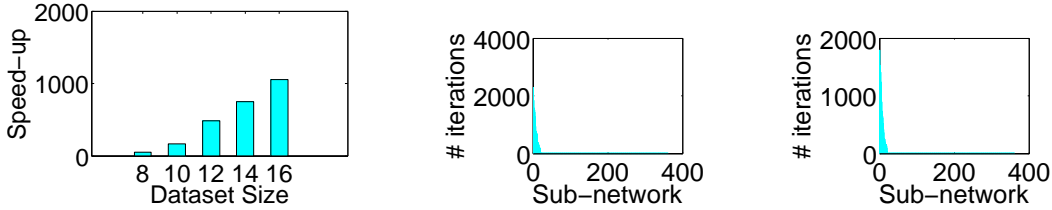

Figure 4: Left: Speed-up of D-EM over EM as a function of dataset size. This is for a chain network with $180$ variables, while observing $50\%$ of the variables. Right Pair: Graphs showing the number of iterations required by each sub-network, sorted descendingly. The problem is for learning Network Pigs while observing $90\%$ of the variables, with convergence based on parameters (left), and on likelihood (right).

at most $2$ distinct examples for sub-network $X_1$, and at most $4$ distinct examples for sub-network $X_{i-1} \to X_i \to X_{i+1}$ (variable $X_i$ is hidden, while variables $X_{i-1}$ and $X_{i+1}$ are observed). Hence, if sub-network $i$ takes $m_i$ iterations to converge, then the inference engine would need to be called at most $2m_1 + 4(m_2 + m_4 + \ldots + m_{n-2})$ times. We will later show that $m_i$ is generally significantly smaller than $m$. Hence, with decomposed learning, the number of calls to the inference engine can be significantly smaller, which can contribute significantly to the obtained savings. [2]

Our analysis suggests that the savings obtained from decomposing the learning problem would amplify as the dataset gets larger. This can be seen clearly in Figure 4 (left), which shows that the speed-up of D-EM over EM grows linearly with the dataset size. Hence, decomposition can be critical when learning with very large datasets.

Interestingly, two of the most prominent (non-commercial) tools for Bayesian networks *do not exhibit this behavior* on the chain network discussed above. This is shown in Figure 5, which compares D-EM to the EM implementations of the GENIE/SMILE and SAMIAM systems,[3] both of which were represented in previous inference evaluations [4]. In particular, we ran these sys-

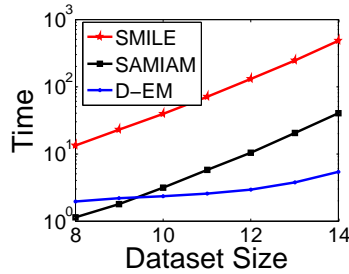

Figure 5: Effect of dataset size (log-scale) on learning time in seconds.

tems on a chain network $X_0 \to \cdots \to X_{100}$, where each variable has $10$ states, and using datasets with alternating observed and hidden variables. Each plot point represents an average over $20$ simulated datasets, where we recorded the time to execute each EM algorithm (excluding the time to read networks and datasets from file, which was negligible compared to learning time).

Clearly, D-EM scales better in terms of time than both SMILE and SAMIAM, as the size of the dataset increases. As explained in the above analysis, the number of calls to the inference engine by D-EM is not necessarily linear in the dataset size. Note here that D-EM used a stricter convergence threshold and obtained better likelihoods, than both SMILE and SAMIAM, in all cases. Yet, D-EM was able to achieve one-to-two orders-of-magnitude speed-ups as the dataset grows in size. On the other hand, SAMIAM was more efficient than SMILE, but got worse likelihoods in all cases, using their default settings (the same seed was used for all algorithms).

**Our second analytic explanation for the obtained savings is based on understanding the dynamics of the convergence test,** used by iterative algorithms such as EM. Such algorithms employ a convergence test based on either parameter or likelihood change. According to the first test, one compares the parameter estimates obtained at iteration $i$ of the algorithm to those obtained at itera-

tion $i-1$. If the estimates are close enough, the algorithm converges. The likelihood test is similar, except that the likelihood of estimates is compared across iterations. In our experiments, we used a convergence test based on parameter change. In particular, when the absolute change in every parameter falls below the set threshold of $10^{-4}$, convergence is declared by EM.

When learning with decomposition, each sub-network is allowed to converge independently, which can contribute significantly to the obtained savings. In particular, with enough observed variables, we have realized that the vast majority of sub-networks converge very quickly, sometimes in one iteration (when the projected dataset is complete). In fact, due to this phenomenon, the convergence threshold for sub-networks can be further tightened without adversely affecting the total running time. In our experiments, we used a threshold of $10^{-5}$ for D-EM, which is tighter than the threshold used for EM. Figure 4 (right pair) illustrates decomposed convergence, by showing the number of iterations required by each sub-network to converge, sorted decreasingly, with convergence test based on parameters (left) and likelihood (right). The vast majority of sub-networks converged very quickly. Here, convergence was declared when the change in parameters or log-likelihood, respectively, fell below the set threshold of $10^{-5}$.

# 6   Related Work

The decomposition techniques we discussed in this paper have long been utilized in the context of inference, but apparently not in learning. In particular, leaf nodes that do not appear in evidence **e** have been called *Barren nodes* in [14], which showed the soundness of their removal during inference with evidence **e**. Similarly, deleting edges outgoing from evidence nodes has been called *evidence absorption* and its soundness was shown in [15]. Interestingly enough, both of these techniques are employed by the inference engines of SAMIAM and SMILE,[4] even though neither seem to employ them when learning network parameters as we propose here (see earlier experiments). When employed during inference, these techniques simplify the network to *reduce the time* needed to compute queries (e.g., conditional marginals which are needed by learning algorithms). However, when employed in the context of learning, these techniques *reduce the number of calls* that need to be made to an inference engine. The difference is therefore fundamental, and the effects of the techniques are orthogonal. In fact, the inference engine we used in our experiments does employ decomposition techniques. Yet, we were still able to obtain orders-of-magnitude speed-ups when decomposing the learning problem. On the other hand, our proposed decomposition techniques do not apply fully to Markov random fields (MRFs) as the partition function cannot be decomposed, even when the data is complete (evaluating the partition function is independent of the data). However, distributed learning algorithms have been proposed in the literature. For example, the recently proposed LAP algorithm is a consistent estimator for MRFs under complete data [10]. A similar method to LAP was independently introduced by [9] in the context of Gaussian graphical models.

# 7   Conclusion

We proposed a technique for decomposing the problem of learning Bayesian network parameters into independent learning problems. The technique applies to incomplete datasets and is based on exploiting variables that are either hidden or observed. Our empirical results suggest that orders-of-magnitude speed-up can be obtained from this decomposition technique, when enough or particular variables are hidden or observed in the dataset. The proposed decomposition technique is orthogonal to the one used for optimizing inference as one reduces the *time* of inference queries, while the other reduces the *number* of such queries. The latter effect is due to decomposing the dataset and the convergence test. The decomposition process incurs little overhead as it can be performed in time that is linear in the structure size and dataset size. Hence, given the potential savings it may lead to, it appears that one must always try to decompose before learning network parameters.

**Acknowledgments**

This work has been partially supported by ONR grant #N00014-12-1-0423 and NSF grant #IIS-1118122.

## Footnotes

[1]The sum-to-one constraints that underlie each component likelihood also partition the sum-to-one constraints of the likelihood function.

[2]The analysis in this section was restricted to chains to make the discussion concrete. This analysis, however, can be generalized to arbitrary networks if enough variables are observed in the corresponding dataset.

[3]Available at `http://genie.sis.pitt.edu/` and `http://reasoning.cs.ucla.edu/samiam/`. SMILE's C++ API was used to run EM, using default options, except we suppressed the randomized parameters option. SAMIAM's Java API was used to run EM (via the CodeBandit feature), also using default options, and the Hugin algorithm as the underlying inference engine.

[4]SMILE actually employs a more advanced technique known as relevance reasoning [8].

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
