[Supplementary Material]

# Decomposing Parameter Estimation Problems

**Khaled S. Refaat, Arthur Choi, Adnan Darwiche**
Computer Science Department
University of California, Los Angeles
{krefaat,aychoi,darwiche}@cs.ucla.edu

**Proposition 1** *The likelihood function $L(\theta|\mathcal{D})$ does not depend on the parameters of variable $X$ if $X$ is hidden in dataset $\mathcal{D}$ and is a leaf of the network structure.*

**Proof** If $\mathbf{d}_i$ is an example of dataset $\mathcal{D}$, then $Pr_\theta(\mathbf{d}_i)$ does not depend on the parameters of variable $X$; see [1, Chapter 6]. Hence, the likelihood function $L(\theta|\mathcal{D}) = \prod_{i=1}^N Pr_\theta(\mathbf{d}_i)$ does not depend on the parameters of variable $X$. □

## 1 Soundness

### 1.1 Decomposing the Likelihood Function

**Theorem 1** *Let $\mathbf{S}$ be a component of $G|\mathbf{O}$ and let $\mathbf{R}$ be the remaining variables of network $G$. If variables $\mathbf{O}$ are observed in example $\mathbf{d}$, we have*

$$Pr_\theta(\mathbf{d}) = \left[\sum_{\Theta_\mathbf{S}^\mathbf{d}} \Theta_\mathbf{S}^\mathbf{d}\right] \left[\sum_{\Theta_\mathbf{R}^\mathbf{d}} \Theta_\mathbf{R}^\mathbf{d}\right].$$

**Proof** Let $\mathbf{N} = \mathbf{S} \cup \mathbf{R}$ be all network variables. One can show that the product $\Theta_\mathbf{S}^\mathbf{d}\Theta_\mathbf{R}^\mathbf{d}$ is a parameter term for $\mathbf{N}$ and $\mathbf{d}$. Moreover, one can show that every parameter term for $\mathbf{N}$ and $\mathbf{d}$ can be written as $\Theta_\mathbf{S}^\mathbf{d}\Theta_\mathbf{R}^\mathbf{d}$. The key observation here is that if variable $X$ is shared by some parameter in $\Theta_\mathbf{S}^\mathbf{d}$ and some parameter in $\Theta_\mathbf{R}^\mathbf{d}$, then $X \in \mathbf{O}$ and its value must be set by example $\mathbf{d}$. Hence, the parameters of $\Theta_\mathbf{S}^\mathbf{d}$ and those of $\Theta_\mathbf{R}^\mathbf{d}$ must be compatible. Hence, one can enumerate all parameter terms $\Theta_\mathbf{N}^\mathbf{d}$ by enumerating all products $\Theta_\mathbf{S}^\mathbf{d}\Theta_\mathbf{R}^\mathbf{d}$:

$$Pr_\theta(\mathbf{d}) = \sum_{\Theta_\mathbf{N}^\mathbf{d}} \Theta_\mathbf{N}^\mathbf{d} = \sum_{\Theta_\mathbf{S}^\mathbf{d}} \sum_{\Theta_\mathbf{R}^\mathbf{d}} \Theta_\mathbf{S}^\mathbf{d}\Theta_\mathbf{R}^\mathbf{d} = \left[\sum_{\Theta_\mathbf{S}^\mathbf{d}} \Theta_\mathbf{S}^\mathbf{d}\right] \left[\sum_{\Theta_\mathbf{R}^\mathbf{d}} \Theta_\mathbf{R}^\mathbf{d}\right].$$

□

### 1.2 Optimizing Component Likelihoods

**Theorem 2** *Consider a sub-network $G$ which is induced by component $\mathbf{S}$ and boundary variables $\mathbf{B}$. Let $\theta$ be the parameters of sub-network $G$, and let $\mathcal{D}$ be a dataset for $G$ that observes boundary variables $\mathbf{B}$. Then $\theta^\star$ is a stationary point for the sub-network likelihood, $L(\theta|\mathcal{D})$, only if $\theta^\star:\mathbf{S}$ is a stationary point for the component likelihood $L(\theta:\mathbf{S}|\mathcal{D})$. Moreover, every stationary point for $L(\theta:\mathbf{S}|\mathcal{D})$ is part of some stationary point for $L(\theta|\mathcal{D})$.*

**Proof** By definition of a sub-network, $\mathbf{S}$ must be a component of $G|\mathbf{B}$. Hence, by Theorem 1, $L(\theta|\mathcal{D}) = L(\theta:\mathbf{S}|\mathcal{D})L(\theta:\mathbf{B}|\mathcal{D})$. Since $\mathbf{S}$ and $\mathbf{B}$ partition the variables of sub-network $G$, the parameters in $\theta:\mathbf{S}$ do not overlap with those in $\theta:\mathbf{B}$, and their union accounts for all sub-network parameters, $\theta$. The theorem then follows immediately from Lemma 1. □

Figure 1: Speedup of D-EDML over EDML on chain networks: three chains (180, 380, and 500 variables) (left), and tree networks (63, 127, 255, and 511 variables) (right), with three random datasets per network/observed percentage, and $2^{10}$ examples per dataset.

## 2 Results

Table 1 and Figure 1 show results for EDML.

| Network | Observed % | Speed-up D-EM | Speed-up D-EDML |
|---|---|---|---|
| alarm | 95.0% | 267.67x | 33.93x |
| alarm | 90.0% | 173.47x | 218.09x |
| alarm | 80.0% | 115.4x | 85.1x |
| alarm | 70.0% | 87.67x | 34.06x |
| alarm | 60.0% | 92.65x | 31.83x |
| alarm | 50.0% | 12.09x | 6.42x |
| win95pts | 95.0% | 591.38x | 49.25x |
| win95pts | 90.0% | 112.57x | 43.43x |
| win95pts | 80.0% | 22.41x | 17.97x |
| win95pts | 70.0% | 17.92x | 14.64x |
| win95pts | 60.0% | 4.8x | 8.4x |
| win95pts | 50.0% | 7.99x | 16.7x |
| andes | 95.0% | 155.54x | 162.63x |
| andes | 90.0% | 52.63x | 90.5x |
| andes | 80.0% | 14.27x | 14.75x |
| andes | 70.0% | 2.96x | 6.24x |
| andes | 60.0% | 0.77x | 2.35x |
| andes | 50.0% | 1.01x | 2.47x |
| diagnose | 95.0% | 43.03x | 127.24x |
| diagnose | 90.0% | 17.16x | 49.69x |
| diagnose | 80.0% | 11.86x | 21.32x |
| diagnose | 70.0% | 3.25x | 11.54x |
| diagnose | 60.0% | 3.48x | 8.72x |
| diagnose | 50.0% | 3.73x | 9.79x |
| water | 95.0% | 811.48x | 88.41x |
| water | 90.0% | 110.27x | 70.0x |
| water | 80.0% | 7.23x | 5.34x |
| water | 70.0% | 1.5x | 1.55x |
| water | 60.0% | 2.03x | 1.82x |
| water | 50.0% | 4.4x | 3.79x |
| pigs | 95.0% | 235.63x | 40.7x |
| pigs | 90.0% | 37.61x | 10.77x |
| pigs | 80.0% | 34.19x | 11.17x |
| pigs | 70.0% | 16.23x | 5.18x |
| pigs | 60.0% | 4.1x | 1.82x |
| pigs | 50.0% | 3.16x | 1.69x |

Table 1: D-EM over EM speed-ups and D-EDML over EDML speed-ups on UAI networks. Three random datasets per network/observed percentage with $2^{10}$ examples per dataset.

## A  Decomposing Stationary Points

A stationary point for function $f(x_1, \ldots, x_n)$ is a point $x_1^\star, \ldots, x_n^\star$ at which the gradient of $f(x_1, \ldots, x_n)$ evaluates to zero. That is,

$$\left. \frac{\partial f}{\partial x_i} \right|_{x_i = x_i^\star} = 0 \text{ for } i = 1, \ldots, n.$$

**Lemma 1** *Consider the non-zero function*

$$f(x_1, \ldots, x_n, y_1, \ldots, y_m) = g(x_1, \ldots, x_n) h(y_1, \ldots, y_m).$$

*Then $x_1^\star, \ldots, x_n^\star, y_1^\star, \ldots, y_m^\star$ is a stationary point of $f$ iff $x_1^\star, \ldots, x_n^\star$ is a stationary point of $g$ and $y_1^\star, \ldots, y_m^\star$ is a stationary point of $h$.*

**Proof**  Consider the following elementary identities:

$$
\begin{aligned}
\frac{\partial f}{\partial x_i} &= g(x_1, \ldots, x_n) \frac{\partial h}{\partial x_i} + h(y_1, \ldots, y_m) \frac{\partial g}{\partial x_i} \\
&= h(y_1, \ldots, y_m) \frac{\partial g}{\partial x_i} \\
\frac{\partial f}{\partial y_i} &= g(x_1, \ldots, x_n) \frac{\partial h}{\partial y_i} + h(y_1, \ldots, y_m) \frac{\partial g}{\partial y_i} \\
&= g(x_1, \ldots, x_n) \frac{\partial h}{\partial y_i}.
\end{aligned}
$$

The lemma follows immediately from these identities since function $f$ is non-zero (which implies that $g$ and $h$ are non-zero). $\square$

## References

[1] Adnan Darwiche. *Modeling and Reasoning with Bayesian Networks*. Cambridge University Press, 2009.