[Reviews · NeurIPS 2014]

Submitted by Assigned_Reviewer_2

This paper proposes a method for finding a max likelihood estimate of the parameters of a Bayesian Network (BN) with hidden variables by decomposing the global optimization problem into a set of independent sub-problems. The main motivation for this decomposition is that in a data set the variables are either always hidden, or always observed - i.e. variables aren’t missing at random. By assuming the learning problem has this property, the authors propose a partitioning of the BN’s parameters in the same way that one decomposes the BN parameter estimation problem in the fully observed case. They then demonstrate the benefit of this decomposition empirically by comparing their method to off-the-shelf implementations of the EM algorithm - perhaps the default approach for BN parameter estimation in missing data problems.

Overall, I found this paper to be very well written and easy to follow, with clear and concise definitions and plenty of well placed examples. My two main criticisms of this paper are with its novelty and significance. The two main ideas appear to be: 1) that a BN with fixed structure can be decomposed into smaller, easier to learn sub-problems; and 2) that the dataset can be compressed by maintaining counts for each of these subproblems. The idea of exploiting a network’s structure to expedite inference/learning has been studied in various hidden-variable architectures, such as RBMs and HMMs. The proposed decomposition falls out naturally, for example, in an RBM architecture. In contrast, the proposed method provides no-benefit in the case of an HMM architecture as there is no decomposition. Along these lines, I felt that the claims made in the experimental suggestion were a bit exaggerated. It seems a bit unfair to take on off-the-shelf EM method, which has no knowledge of the fixed pattern of missingness, and comparing to a method designed explicitly to exploit such a missingness pattern. The reason these general purpose methods can’t take advantage of the “data projection” operator is because they do not assume that the missingness pattern is fixed! Rather than just reporting the speed-up, it would have been better to show how the speed-up degrades as the decomposition becomes less prominent. Increasing the percentage of observed variables does not tell us if the number of sub-networks decreases or increases, or anything about the size of the sub-networks relative to the entire network. In addition, how does the method perform as the size of the data set is changed. Finally, I think the authors should have addressed the role of regularization. What type of priors on the hidden parameters are supported by the proposed method and which are not.

Detailed Comments:
- Line 261: what is meant by classical EM? EM runs inference on each training instance, while D-EM runs on a sub-network using summarized/compressed data.
- Line 267: Why does making all unobserved variables hidden make the problem harder?
- Line 270: The caption of Figure 3 suggests that I should see three differently bars, one for each chain length. However, I only see one set of bars.
- Line 284: In Table 1, provide some summary information of the UAI networks, e.g. number of variables, tree-width, variable arity, etc…
- Line 324: In Figure 4, the center and right charts were too small to reveal anything. Perhaps a log-log plot would be better.
Summary: This is a well-written and clear paper, but I think the proposed method is well understood by the graphical models community and is not that original. I also feel that the experiments section was not objective enough - both the strengths and the weakness of a method need to be discussed by the authors.

Submitted by Assigned_Reviewer_36

The paper presents a new technique for speeding up parameter learning in Bayesian networks in presence of latent or hidden variables (known structure case). The key idea is to decompose the learning problem into independent sub-problems by taking advantage of some basic (well-known) structural properties of Bayesian networks (e.g., d-separation). The authors show both empirically and analytically that their technique is quite promising; the empirical speedups are impressive.

Quality: The quality of writing is generally good. Actually this is quite an easy paper to write, I wonder why no one thought about this idea before.

Clarity: The paper is mostly clear. The notation can be simplified greatly. However, it is correct as far as I can tell.

Originality: The idea is original but the paper is not a game changer. The paper can be classified into a category of papers that makes a clever observation which was somehow missed by researchers. The key observation in the paper is nicely summarized in the related work section: "When employed during inference, these techniques simplify the network to reduce the time needed to compute queries (e.g., conditional marginals which are needed by learning algorithms). However, when employed in the context of learning, these techniques reduce the number of calls that need to be made to an inference engine." Here the authors are talking about structural properties of the Bayesian networks.

Significance: In my opinion, the paper is not going to have a significant impact. The techniques presented are pretty much stand-alone and I'm not sure how they can be extended or applied to other parameter learning tasks (e.g., Markov networks; the authors acknowledge that this is not possible).
Summary: Overall, the paper is well-written and presents a simple but clever idea for speeding up parameter learning in Bayesian networks.

Submitted by Assigned_Reviewer_38

This paper proposes a method for speeding up EM for learning Bayesian networks from incomplete data. The method decomposes the likelihood function into smaller independent components by noting that fully unobserved nodes do not occur in the likelihood function, and fully observed nodes create independent components. Although the idea has been used in inference before, it has not been applied in learning, and the paper demonstrates that this technique can be applied to significantly speed up EM. Despite its simplicity, the application of this technique can potentially be very useful to researchers in the field.

The paper is clearly written. Minor comments:
- It will be good to make it clear what is meant by "made all unobserved variables hidden (as opposed to missing at random)" since unobserved, hidden and missing may be treated synonymously.
- Include the network sizes for experiments in Table 1.

Added after rebuttal: I've read and considered the rebuttal.
Summary: This paper proposes speeding up EM for learning Bayesian network from incomplete data by using a technique used in decomposing likelihood for inference. Despite its simplicity, it can be very useful to researchers in the field.
Author Feedback
Author rebuttal: We thank all the reviewers for their valuable time and insightful comments. We address each reviewer in turn.

Assigned_Reviewer_2

1. "The main motivation for this decomposition is that in a data set the variables are either always hidden, or always observed - i.e. variables aren't missing at random. By assuming the learning problem has this property,..."

Please note that our method does not require that variables fall into one of these categories; one can still have variables that are missing at random except that the method will not exploit these. Our experiments, however, assumed that no variables are missing at random as that ends up being more challenging for EM, therefore, emphasizing more the importance of the proposed decomposition method.

2. "It seems a bit unfair to take on off-the-shelf EM method, which has no knowledge of the fixed pattern of missingness, and comparing to a method designed explicitly to exploit such a missingness pattern."

The fixed patterns of missingness that we exploit are readily available from the data. That is, using a single pass through the data, one can detect whether nodes are fully observed, fully un-observed, or partially observed. Our main point is that off-the-shelf EM implementations do not make this pass to detect the patterns and therefore do not exploit them (as we show for SMILE and SamIam). Hence, our point behind these experiments is three fold: standard EM implementations do not appear to implement the technique we propose, implementing the technique incurs little overhead, and the savings led to can be very significant.

3. "Increasing the percentage of observed variables does not tell us if the number of sub-networks decreases or increases, or anything about the size of the sub-networks relative to the entire network."

We noticed empirically that increasing the number of observed variables generally increases the number of sub-networks. In fact, once all leaf variables are observed (no node pruning), increasing the number of observed variables can never decrease the number of sub-networks (this is because observing a variable will lead to removing edges from the network).

4. "In addition, how does the method perform as the size of the data set is changed."

The paper includes a specific experiment on this, showing that the savings obtained by the proposed method gets significantly larger as the dataset gets larger (due to data compression). Please see Figure 4 (left) and Figure 5.

5. "Finally, I think the authors should have addressed the role of regularization."

Our experiments assumed a Dirichlet prior that corresponds to Laplace smoothing. We will elaborate more on this.

6. "...both the strengths and the weakness of a method need to be discussed by the authors."

The only disadvantage of our method is the little overhead associated with the single pass on the dataset, and the single pass on the network, which could be wasted if they lead to no decomposition. Please note that each of these passes can be done in linear time. We will emphasize this point in a paper revision.

-Line 261: Classical EM is the EM that runs decomposed inference given each unique example. We assume reasonable implementations of EM that already decompose the network for the purpose of inference, such as GeNIe/SMILE. D-EM decomposes the learning problem itself, getting decomposed convergence and data compression.
-Line 267: In practice, EM is considered less effective when learning under the presence of fully-unobserved variables, in contrast to just partially-observed variables. We will add the appropriate references.
-Line 270: Every bar is the average.
-Lines 284 and 324: Good suggestions. Thanks!

Assigned_Reviewer_36

The objective function in MRFs consists of two terms: the data term (i.e. the term that depends on the data), and the model term (i.e. the partition function). We have been recently investigating the possibility of applying the same decomposition techniques to the data term, with initial promising results. While this does not decompose the learning problem into independent learning problems (due to the undecomposed model term), it can still compress the data term via data projection, leading to speed-ups, when computing the gradient. When we said "do not apply fully", we actually meant it does apply but not fully. Namely, in Bayesian networks (BNs), we get data decomposition and convergence test decomposition, whereas in MRFs, we only get data decomposition. We hope this will show the potential impact of the proposed method beyond BNs.

Assigned_Reviewer_38

Thank you very much for your encouraging review. We will incorporate all suggestions into the final version, if accepted.